# Adjusting for Dropout Variance in Batch Normalization and Weight Initialization

**Dan Hendrycks**[*]
University of Chicago
dan@ttic.edu

**Kevin Gimpel**
Toyota Technological Institute at Chicago
kgimpel@ttic.edu

## Abstract

We show how to adjust for the variance introduced by dropout with corrections to weight initialization and Batch Normalization, yielding higher accuracy. Though dropout can preserve the expected input to a neuron between train and test, the variance of the input differs. We thus propose a new weight initialization by correcting for the influence of dropout rates and an arbitrary nonlinearity's influence on variance through simple corrective scalars. Since Batch Normalization trained with dropout estimates the variance of a layer's incoming distribution with some inputs dropped, the variance also differs between train and test. After training a network with Batch Normalization and dropout, we simply update Batch Normalization's variance moving averages with dropout off and obtain state of the art on CIFAR-10 and CIFAR-100 without data augmentation.

## 1 Introduction

Weight initialization and Batch Normalization greatly influence a neural network's ability to learn. Both methods can allow for a unit-variance neuron input distribution. This is desirable because variance larger or smaller than one may cause activation outputs to explode or vanish. In order to encourage unit-variance, early weight initialization attempts sought to adjust for a neuron's fan-in (LeCun et al., 1998). More recent initializations correct for a neuron's fan-out (Glorot & Bengio, 2010). Meanwhile, some weight initializations compensate for the compressiveness of the ReLU nonlinearity (the ReLU's tendency to reduce output variance) (He et al., 2015). Indeed, He et al. (2015) also show that initializations without a specific, small corrective factor can render a neural network untrainable. To address this issue Batch Normalization (Ioffe & Szegedy, 2015) reduces the role of weight initialization at the cost of up to 30% more computation (Mishkin & Matas, 2015). A less computationally expensive solution is the LSUV weight initialization, yet this still requires computing batch statistics, a special forward pass, and makes no adjustment for backpropagation error signal variance (Mishkin & Matas, 2015). Similarly, weight normalization uses a special feed-forward pass and computes batch statistics (Salimans & Kingma, 2016). The continued development of variance stabilizing techniques testifies to its importance for neural networks.

Both Batch Normalization and previous weight initializations do not accommodate the variance introduced by dropout, and we contribute methods to fix this. First we demonstrate a new weight initialization technique which includes a new correction factor for a layer's dropout rate and adjusts for an arbitrary nonlinearity's effect on the neuron output variance. All of this is obtained without computing batch statistics or special adjustments to the forward pass, unlike recent methods to control variance (Ioffe & Szegedy, 2015; Mishkin & Matas, 2015; Salimans & Kingma, 2016). By this new initialization, we enable faster and more accurate convergence. Afterward, we show that networks trained with Batch Normalization can improve their accuracy by adjusting for dropout's variance. We accomplish this by training a network with both Batch Normalization and dropout, then after training we feed forward the training dataset with dropout off to reestimate the Batch Normalization variance estimates. Because of this simple, general technique, we obtain state of the art on CIFAR-10 and CIFAR-100 without data augmentation.

---

[*]Work done while the author was at TTIC. Code available at github.com/hendrycks/init

## 2 WEIGHT INITIALIZATION

### 2.1 DERIVATION

In this section, we derive our new initialization by considering a neuron input distribution and its major sources of variance. We accomplish this by separately considering the feedforward and the backpropagation stages.

### 2.1.1 THE FORWARD PASS

We use $f$ to denote the pointwise nonlinearity in each neural network layer. For simplicity, we use the term "neuron" to refer to an entry in a layer before applying $f$. Let us also call the input of the $l$-th layer $z^{l-1}$, and let the $n_{\text{in}} \times n_{\text{out}}$ weight matrix $W^l$ map from layer $l-1$ to $l$. Let an entry of this matrix be $w^l$. In our upcoming initialization, we initialize each column of $W^l$ on the unit hypersphere so that each column has an $\ell_2$ norm of 1. Now, if we assume that this network is trained with a dropout *keep* rate of $p$, we must scale the output of a layer by $1/p$. Also assume $f(z^{l-1})$ and $W$ are zero-centered. With that now specified, we conclude that neuron $i$ of layer $z^l$ has the variance

$$\text{Var}(z_i^l) = \text{Var}\left(\sum_{k=1}^{n_{\text{in}}} W_{ki}^l f(z^{l-1})_k / p\right)$$

$$\approx \frac{n_{\text{in}}p}{p^2}\text{Var}(w^l f(z^{l-1}))$$

$$= \mathbb{E}[f(z^{l-1})^2]/p$$

because $\text{Var}(w^l) = 1/n_{\text{in}}$, since we initialized $W^l$'s columns on the unit hypersphere. Knowing this variance allows us to adjust for the influence of an *arbitrary* nonlinearity and a desired dropout rate.

We empirically verify that a weight initialization with this forward correction allows for consistent input distribution variance throughout the layers of a 20-layer fully connected network for differing dropout rates. The first 15 layers have 500 neurons, and the last 5 layers have 250 neurons. Specifically, we can encourage unit variance by dividing $W$, initialized on the unit hypersphere, by $\sqrt{\mathbb{E}(f(z^{l-1})^2)/p}$. Let us compare this correction to other initializations by feeding forward a random standard normal matrix through 20 layers. Figure 1 shows the results of such an experiment, and in the experiment we use a ReLU activation function. Of course, as He initialization was designed specifically for the ReLU, it performs well when $p = 1$, but has an exploding distribution when there is dropout. Only the initialization with a $\sqrt{\mathbb{E}(f(z^{l-1})^2)/p}$ corrective term demonstrates stability when a feedforward does or does not use dropout.

### 2.1.2 BACKPROPAGATION

A similar analysis shows that if $L$ is our loss function and $\delta^l = \frac{\partial L}{\partial z^l}$, then

$$\text{Var}(\delta^l) \approx pn_{\text{out}}\text{Var}(w^{l+1}\delta^{l+1}f'(z^l)) = p\mathbb{E}[f'(z^l)^2].$$

In appendix A we empirically verify that this backward correction allows for consistent backpropagation error signal variance throughout the layers of a 20-layer network for differing dropout rates.

### 2.2 OUR INITIALIZATION

We want that $\text{Var}(z^l) = 1$ and $\text{Var}(\delta^l) = \text{Var}(\delta^{l+1})$. To meet these different goals, we can initialize our weights by *adding* these variances, while others take the arithmetic mean of these variances or ignore the backpropagation variance altogether (He et al., 2015; Glorot & Bengio, 2010). Therefore, if $W^l$ has its columns sampled uniformly from the surface of a unit hypersphere or is an orthonormal matrix, then our initialization is

$$W^l/\sqrt{\mathbb{E}[f(z^{l-1})^2]/p + p\mathbb{E}[f'(z^l)^2]}.$$

For convolutional neural networks, adjusting for the backpropagation signal is less common, so one could simply use the initialization $W^l/\sqrt{\mathbb{E}[f(z^{l-1})^2]/p}$. This initialization accounts for the influence of dropout rates and an arbitrary nonlinearity. We can simply initialize a random standard

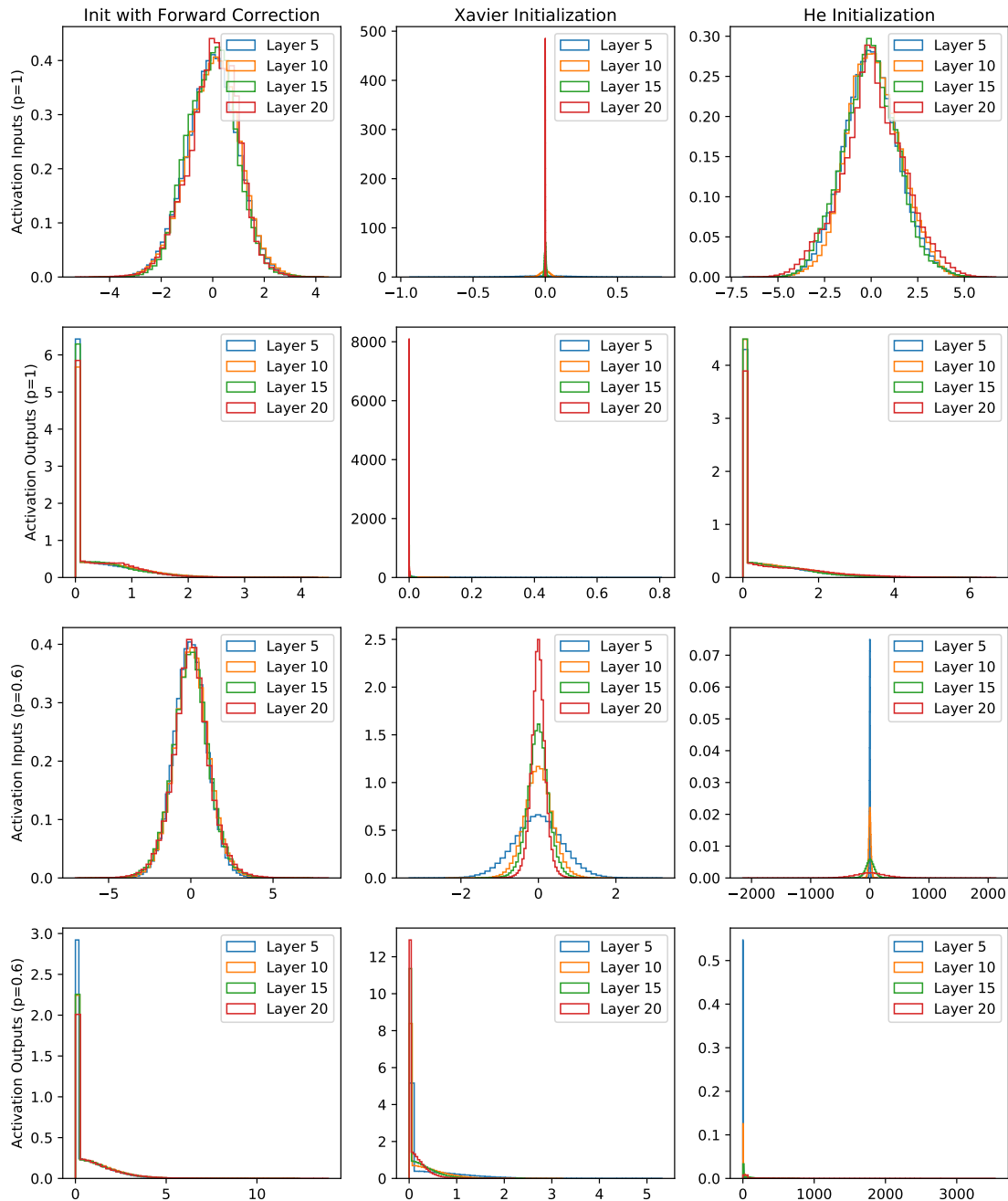

Figure 1: A comparison of a unit hypersphere initialization with a forward correction, the Xavier initialization, and the He initialization. Each plot shows the probability density function of a neuron's inputs and outputs across layers. In particular, the range of values vary widely between each initialization, with exponential blowups and decay for He and Xavier initializations, respectively. Values set to zero by dropout are removed from the probability density functions.

| Activation | $\mathbb{E}(f(z^{l-1})^2)$ | $\mathbb{E}(f'(z^l)^2)$ |
|---|---|---|
| Identity | 1 | 1 |
| ReLU | 0.5 | 0.5 |
| GELU ($\mu = 0, \sigma = 1$) | 0.425 | 0.444 |
| tanh | 0.394 | 0.216 |
| ELU ($\alpha = 1$) | 0.645 | 0.671 |

Table 1: Activation adjustment estimates for $z^{l-1}, z^l$ following a standard normal distribution.

Gaussian matrix and normalize its last dimension to generate $W^l$. Another strength of this initialization is that the expectations are similar to the values in Table 1 for standardized input data, so computing mini-batch statistics is needless for our initialization (Hendrycks & Gimpel, 2016). We need only substitute in appropriate scalars during initialization. Let us now see these new adjustments in action.

## 2.3 EXPERIMENTS

In the experiments that follow, we utilize the MNIST dataset, a 10-class grayscale image dataset of handwritten digits with 60k training examples and 10k test examples. Then we consider CIFAR-10 (Krizhevsky, 2009), a 10-class color image dataset with 50k training examples and 10k test examples. We use these data to compare our initialization with Xavier and He initializations on a fully connected neural network and with the He initialization on a convolutional neural network.

### 2.3.1 MNIST

Let us verify that our initialization competes with previous weight initialization schemes. To this end, we train a fully connected neural network with ReLUs, ELUs ($\alpha = 1$), and the tanh activation (Clevert et al., 2016). Each 8-layer, 256 neuron wide neural network is trained for 25 epochs with a batch size of 64. From the training set, 5000 examples are held out for a validation set. With the validation set, we tune over the learning rates $\{10^{-3}, 10^{-4}, 10^{-5}\}$ and 7 other learning rates randomly chosen from $[10^{-1}, 10^{-5}]$. We optimize with Adam (Kingma & Ba, 2015). We also perform this task with no dropout, a dropout keep rate of $0.5$, and a dropout keep rate of $0.3$. Figure 2 indicates that our initialization provides faster convergence at a dropout keep rate of $0.5$ for activations like the ReLU and great gains when the dropout keep rate decreases further.

### 2.3.2 CIFAR-10

Since VGG Net architectures (Simonyan & Zisserman, 2015) require considerable regularization and careful initialization, we use a highly regularized variant (Zagoruyko, 2015) of the architecture for our next initialization experiment. The VGG Net-like network has the stacks $(2 \times 3 \times 64), (2 \times 3 \times 128), (3 \times 3 \times 256), (3 \times 3 \times 512), (3 \times 3 \times 512)$ followed by two fully-connected layers, each with 512 neurons. To regularize the deep network, we keep 70% of the neurons in the first layer, 60% in layer 3, 60% in the first two layers of the last three stacks, and 50% in the fully connected layers. Max pooling occurs after every stack, ReLU activations are applied on every neuron, and we $\ell_2$ regularize with a strength of $5 \times 10^{-4}$. Layer width, filter count, $\ell_2$ regularization strength, and dropout rate hyperparmeters are from (Zagoruyko, 2015). We compare our initialization (while deactivating the back-

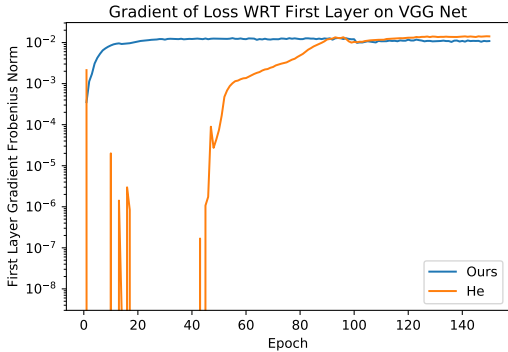

Figure 3: Our weight initialization enables consistent, healthy gradients and a worse initialization may require dozens of epochs to begin optimizing the first layer due to vanishing gradients.

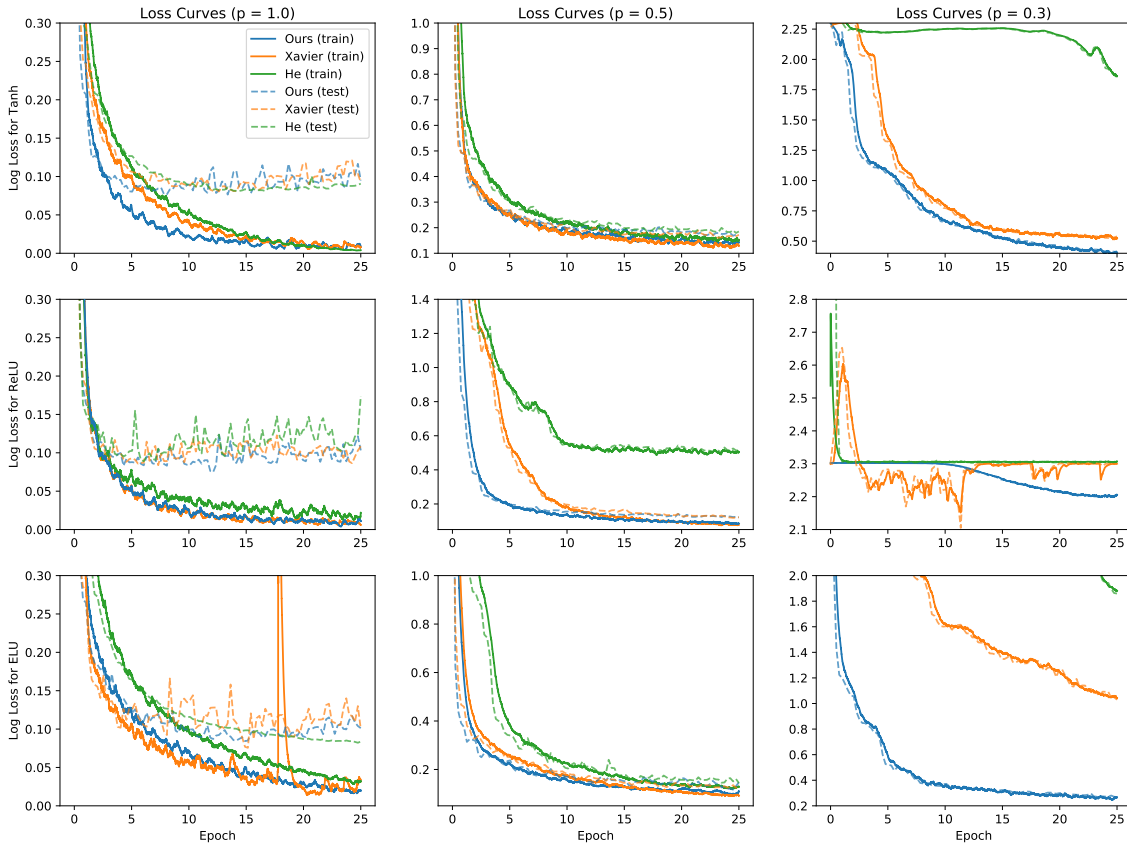

Figure 2: MNIST Classification Results. The first row shows the log loss curves for the $\tanh$ unit, the second row is for the ReLU, and the third the ELU. The leftmost column shows loss curves when there is no dropout, the middle when the dropout rate is 0.5, and rightmost is when the dropout preservation probability rate is 0.3. Each curve is selected by tuning over learning rates.

propagation variance term) and the He initialization. Since the Xavier initialization is not as prominent in convolutional neural networks we do not test it. We optimize this network with two different optimizers. First, we use Nesterov momentum and tune with 10 learning rates, with 7 chosen randomly from $[10^{-1}, 10^{-5}]$ and three deterministically chosen from $\{10^{-2}, 10^{-3}, 10^{-4}\}$. In separate runs, we train with the Adam optimizer and tune with 10 learning rates, with 7 chosen randomly from $[10^{-1}, 10^{-5}]$ and three deterministically chosen from $\{10^{-3}, 10^{-4}, 10^{-5}\}$. With both optimizers we decay the learning rate by 0.1 every at the 100th and 125th epoch all while training for 150 epochs. The results in Figure 4 demonstrate the importance of small corrective dropout factors because the factors' influence on neuron input variance changes exponentially as the network depth increases, and this can lead to vanishing update signals as shown in Figure 3. We should note that the He initialization rendered the network untrainable for more learning rates like when the learning rate was 0.01 with Nesterov momentum. However, the network converged with our initialization at this learning rate. Ultimately, our initialization provided more consistent, quick, and accurate convergence. With the Adam Optimizer, the VGG Net obtained 9.51% test set error under our initialization and 10.54% with the He initialization. Last, we use Nesterov momentum, which is a more common optimizer for deep convolutional neural networks. With Nesterov momentum, we obtained **7.41**% error with our initialization and 24.65% with the He initialization.

## 3   BATCH NORMALIZATION VARIANCE RE-ESTIMATION

Batch Normalization aims to prevent an exploding or vanishing feedforward signal just like a good weight initialization. However, Batch Normalization has its own caveats. A practical concern,

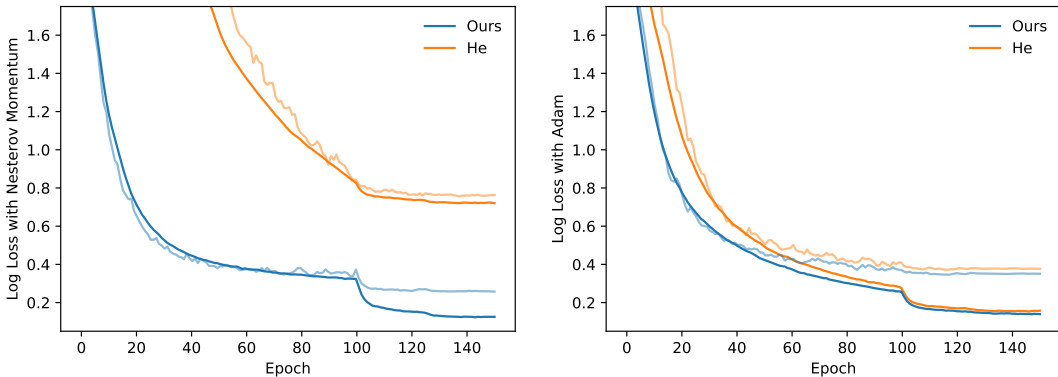

Figure 4: CIFAR-10 VGG Net Results. The left convergence curves show how the network trained with Nesterov momentum, and the right shows the training under the Adam optimizer. Training set log losses are the darker curves, and the fainter curves are the test set log loss curves.

voiced in Mishkin & Matas (2015), is that Batch Normalization can increase the feedforward time by up to 30%. Also, Ba et al. (2016) remind us that Batch Normalization cannot be applied to tasks with small batch sizes or online learning tasks lest we normalize a batch based upon mean and variance estimates from a small or single example. Batch Normalization can be used to stabilize the feedforward signal of a network with dropout, removing the need for a weight initialization which corrects for dropout variance. However, correcting for dropout's variance is still necessary. It so happens that Batch Normalization requires that its own variance estimates be corrected before testing.

In this section we empirically show that Batch Normalization *with dropout* also requires special care because Batch Normalization's estimated variance should differ between train and test. In our weight initialization derivation section 2.1, we saw that the variance of a neuron's input grows when dropout is active. Consequently, Batch Normalization's variance estimates are greater when dropout is active because a neuron's input variance is greater with dropout. But dropout is deactivated during testing, and the variance estimates Batch Normalization normally uses are accurate when dropout is active not inactive. For this reason, we re-estimate the Batch Normalization variance parameters after training. We accomplish this by simply feeding forward the training data with dropout deactivated and only letting the Batch Normalization variance running averages update. When re-estimating the variance no backpropagation occurs. Let us now verify that re-estimating the Batch Normalization variance values improves test performance.

In our experiment, we turn our attention to state of the art convolutional neural networks as they use Batch Normalization and dropout. For example, Densely Connected Networks (DenseNets) (Huang et al., 2016) use dropout with Batch Normalization when training without data augmentation. Training without data augmentation is of interest because it demonstrates how data efficient an architecture is, and some images have their meaning destroyed under augmentation like mirroring (e.g., the mirror image of the digit "7" is meaningless). Moreover, Zagoruyko & Komodakis (2016) use dropout even when there is data augmentation as Batch Normalization alone does not sufficiently regularize the network. Please note that in this experiment we are only testing the effect of Batch Normalization variance adjustments, and we are not testing the effect of different weight initializations.

We turn to DenseNets in this experiment because, to our knowledge, they hold the state of the art on CIFAR-10 and CIFAR-100 without data augmentation. We train a DenseNet with dropout and Batch Normalization and re-estimate the Batch Normalization variance parameters outside of training to achieve large error reductions. These DenseNets are trained just as described in the original paper *except* that every 5 epochs we reset the momentum variable following a discussion with a DenseNet paper author, as this might improve accuracy. We save the DenseNet model when it has trained for half of the scheduled epochs (when it is "Halfway") and when it is entirely done training. Then, using these models, we feed forward the training data with *dropout off* for one epoch

without performing any backpropagation. While the data feeds forward, we only allow the Batch Normalization moving average estimate of the variance to update. In no way does this variance re-estimation at the Halfway point affect future training because we do not train with these re-estimated variance parameters. Now, DenseNets hold the state of the art on CIFAR-10 and CIFAR-100 without data augmentation. Specifically, they obtain 5.77% error on CIFAR-10 without data augmentation and 23.42% on CIFAR-100 without data augmentation. Table 2 shows the results of Batch Normalization variance moving average re-estimation. The figure shows $L, k$, and $p$, which are the number of layers $L$, the growth factor $k$, and the dropout keep probability $p$. As an example of a table row, SVHN Original shows the error achieved in the original DenseNet paper. The row below shows the DenseNet we trained at the Halfway point ("Halfway Error") and at the end of training ("Error"), and the error decreased under re-estimating the Batch Normalization variance. The effect of updating the variance estimation is shown under columns with "BN Update." We see that simply feeding forward the training data without dropout and allowing the Batch Normalization variance moving averages to update lets us surpass the state of the art on CIFAR-10 and CIFAR-100 and can sometimes improve accuracy by more than 2%.

| Dataset (Architecture) | Halfway Error | Halfway Error w/ BN Update | Error | Error w/ BN Update |
|---|---|---|---|---|
| SVHN ($L = 40, k = 12, p = 0.8$) Original | — | — | 1.79 | — |
| SVHN ($L = 40, k = 12, p = 0.8$) Ours | 5.18 | 4.19 | 1.92 | 1.85 |
| CIFAR-10 ($L = 100, k = 12, p = 0.8$) Original | — | — | 5.77 | — |
| CIFAR-10 ($L = 100, k = 12, p = 0.8$) Ours | 6.37 | 6.07 | **5.62** | **5.38** |
| CIFAR-100 ($L = 100, k = 24, p = 0.8$) Original | — | — | 23.42 | — |
| CIFAR-100 ($L = 100, k = 12, p = 0.8$) Original | — | — | 23.79 | — |
| CIFAR-100 ($L = 100, k = 12, p = 0.7$) Ours | 24.56 | **22.48** | 23.91 | **22.48** |
| CIFAR-100 ($L = 100, k = 12, p = 0.8$) Ours | 23.89 | **22.86** | 22.65 | **22.17** |

Table 2: DenseNet Results with Batch Normalization Variance Re-Estimation. DenseNets without any Batch Normalization variance re-estimation are shown in "Halfway Error" and "Error" columns. Rows with "Original" denote values from Huang et al. (2016). Bold values indicate that the previous state of the art is exceeded. For CIFAR-10 without data augmentation the previous state of the art was $5.77\%$ and for CIFAR-100 without data augmentation it was $23.42\%$.

## 4  DISCUSSION

In practice, if we lack an estimate for a nonlinearity adjustment factor, then 0.5 is a reasonable default. A justification for a 0.5 adjustment factor comes from connections to previous weight initializations. This is because if $p = 1$ and we default the adjustments to 0.5, our initialization is the "Xavier" initialization if we use vectors from within the unit hypercube rather than vectors on the unit hypersphere (Glorot & Bengio, 2010). Knowing this connection, we can therefore generalize Xavier initialization to

$$\mathrm{Unif}[-1, 1] \times \sqrt{3} \left/ \sqrt{\frac{n_{\mathrm{in}}}{p} \mathbb{E}[f(z^{l-1})^2] + pn_{\mathrm{out}}\mathbb{E}[f'(z^l)^2]} \right. .$$

Furthermore, we can optionally exclude the backpropagation variance term—in this case, if $p = 1$ and $f$ is a ReLU, our initialization is He's initialization if we use random normal weights (He et al., 2015). Note that since He et al. (2015) considered a $0.5$ corrective factor to account for the ReLU's compressiveness (its tendency to reduce output variance), it is plausible that $\mathbb{E}[f(z^{l-1})^2]$ is a general adjustment for a nonlinearity's compressiveness. Since most neural network nonlinearities are compressive, 0.5 is a reasonable default adjustment.[1] Also recall that our initialization with the backpropagation variance term amounts is

$$W^l \left/ \sqrt{\mathbb{E}[f(z^{l-1})^2]/p + p\mathbb{E}[f'(z^l)^2]} \right. .$$

---

[1]Note that the first hidden layer is adjacent to neurons which can be viewed as having an identity activation. For these, a 1.0 factor is more appropriate, but the practical difference is miniscule.

If we use the 0.5 corrective factor default and we do not apply any dropout, then we are left with $W^l$, an orthonormal matrix or a matrix with its columns on a unit hypersphere.

## 5 CONCLUSION

A simple modification to previous weight initializations shows marked improvements on fully connected and convolutional architectures. Unlike recent variance stabilization techniques, ours only relies on simple corrective factors and not special forward passes or batch statistics. For highly-regularized networks, the convergence gains are conspicuous and networks without the corrective factors are harder to train. Therefore, if a user wants to train online or not pay the computational cost Batch Normalization imposes, he or she would do well to apply a dropout corrective factor to their weight initialization matrix.

If a user is able to use Batch Normalization, the effect of dropout still cannot be ignored. The Batch Normalization variance moving averages differ between train and test, so when training is complete, we re-estimate those variance parameters. This is accomplished by feeding the training data forward for one epoch without dropout on and only allowing the variance moving averages to change. By doing so, networks can improve their accuracy notably. Indeed, by applying this simple, highly general technique, we achieved the state of the art on CIFAR-10 and CIFAR-100 without data augmentation.

## ACKNOWLEDGMENTS

We would like to thank Eric Martin for numerous suggestions, Steven Basart for training the SVHN DenseNet, and our anonymous reviewers for suggestions. We would also like to thank NVIDIA Corporation for donating several TITAN X GPUs used in this research.

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

## A A RANDOM BACKPROPAGATION

We can "feed backward" a random Gaussian matrix with standard deviation $0.01$ and see how different initializations affect the distribution of error signals for each layer. Figure 5 shows the results when the backward correction factor is $p\mathbb{E}[f'(z^l)^2]$. Again, we used the ReLU activation function due to its widespread use, so the He initialization performs considerably well.

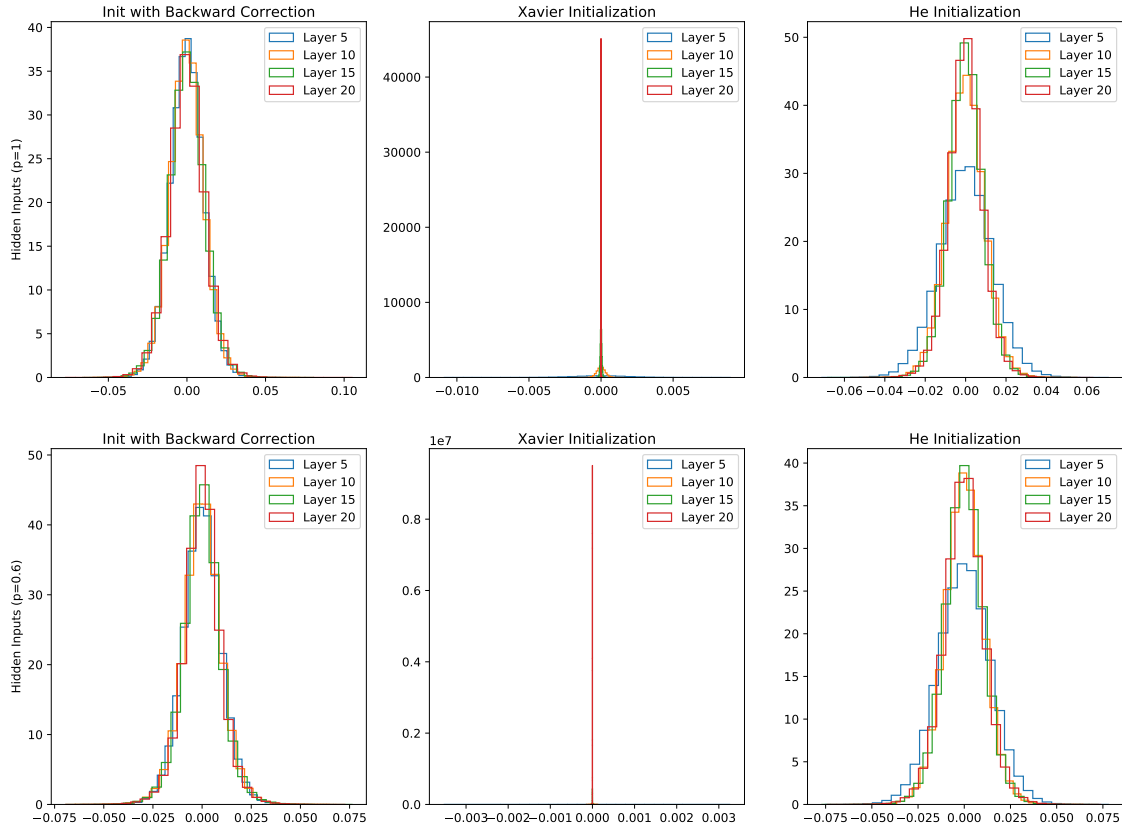

Figure 5: A comparison of a unit hypersphere initialization with a backward correction, the Xavier initialization, and the He initialization. Values set to zero by dropout are removed from the normalized histograms. The first row has a dropout keep probability of $1.0$, the second $0.6$.

