# Peer review of "Adjusting for Dropout Variance in Batch Normalization and Weight Initialization"

_ICLR 2017 — rejected_

[Reviewer Comment · AnonReviewer2 · 03 Dec 2016]
**about the setup and comparisons**

-Why don't you compare to also recent state of art methods like resnet variants or denseNet  for Section 2?
- The parameters are set to fix values of selected from small set of values. However tuning with random search or bayesian optimization is the two common ways to obtain meaningful comparisons. At this stage, it is hard to see whether the difference is coming from proposed approaches since the parameters are not fine tuned reasonably.

- Are there any results on Imagenet?

[Official Review · AnonReviewer1 · rating 6 · confidence 4 · 15 Dec 2016]
**Interesting initialization approach together with a simple inference trick to improve accuracy. Due to limitied experimental validation and theoritical analysis, hard to judge the contribution.**

The paper presents an approach for compensating the input/activation variance introduced by dropout in a network. Additionally, a practical inference trick of re-estimating the batch normalization parameters with dropout turned off before testing. 

The authors very well show how dropout influences the input/activation variance and then scale the initial weights accordingly to achieve unit variance which helps in avoiding activation outputs exploding or vanishing. It is shown that the presented approach serves as a good initialization technique for deep networks and results in performances op par or slightly better than the existing approaches. The limited experimental validation and only small difference in accuracies compared to existing methods makes it difficult to judge the effectiveness of presented approach. Perhaps observing the statistics of output activations and gradients over training epochs in multiple experiments can better support the argument of stability of the network using proposed approach.

Authors might consider adding some validation for considering the backpropagation variance. On multiple occasions comparison is drawn against batch normalization which I believe does much more than a weight initialization technique. The presented approach is a good initialization technique just not sure if its better than existing ones.

[Official Review · AnonReviewer3 · rating 7 · confidence 4 · 16 Dec 2016]
**Interesting observation about the usefulness of adjusting for dropout variance that people should know**

The main observation made in the paper is that the use of dropout increases the variance of neurons. Correcting for this increase in variance, in the parameter initialization, and in the test-time statistics of batch normalization, improves performance, as is shown reasonably convincingly in the experiments.

This observation is important, as it applies to many of the models used in the literature. It's not extremely novel (it's been observed in the literature before that our simple dropout approximations at test time do not achieve the accuracy obtained by full Monte Carlo dropout)

The paper could use more experimental validation. Specifically:

- I'm guessing the correction for dropout variance at test time is not only specific to batch normalization: Standard dropout, in networks without batch normalization, corrects only for the mean at test time (by dividing activations by one minus the dropout probability). This work suggests it would be beneficial to also correct for the variance. Has this been tested?

-  How does the dropout variance correction compare to using Monte Carlo dropout at test time? (i.e. just averaging over a large number of random dropout masks)

[Official Review · AnonReviewer2 · rating 5 · confidence 4 · 20 Dec 2016]
**results are not convincing**

This paper proposes new initialization for particular architectures and a correction trick to batch normalization to correct variance introduced by dropout. While authors state interesting observations, the claims are not supported with convincing results.

I guess Figure 1 is only for mnist and for only two values of p with one particular network architecture, the dataset and empirical setup is not clear.

The convergence is demonstrated only for three dropout values in Figure 2 which may cause an unfair comparison. For instance how does the convergence compare for the best dropout rate after cross-validation (three figures each figure has three results for one method with different dropouts [bests cv result for each one])? Also how is the corresponding validation error and test iterations?  Also only mnist does not have to generalize to other benchmarks.

Figure 3 gives closer results for Adam optimizer, learning rate is not selected with random search or bayesian optimization, learning decay iterations fixed and regularization coefficient is set to a small value without tuning. A slightly better tuning of parameters may close the current gap. Also Nesterov based competitor gives unreasonably worse accuracy compared to recent results which may indicate that this experiment should not be taken into account. 

In Table 2, there is no significant improvement on CIFAR10. The CIFAR100 difference is not significant without including batch normalization variance re-estimation. However there is no result for 'original with BN update' therefore it is not clear whether the BN update helps in general or not. SVHN also does not have result for original with BN update.

There should be baselines with batch normalizations for Figure 1,2 3 to support the claims convincingly.  The main criticism about batch normalization is additional computational cost by giving (Mishkin et al, 2016 ) as reference however this should not be a reason to not to compare the initialization to batch-normalization.  In fact, (Mishkin et al, 2016) performs comparison to batch normalization and also with and without data augmentation with recent state of art architectures.

None of the empirical results have data augmentation. It is not clear if the initialization or  batch normalization update will help or make it worse for that case.

Recent state of art methods methods like Res Net variant and Dense Net scale to many depths and report result for ImageNet. Although the authors claim that this can be extended to residual network variants, it is not clear if there is going to be any empirical gain for that architectures.  

This work requires a comprehensive and fair comparison. Otherwise the contribution is not significant.

[Author Response · Dan Hendrycks · 14 Jan 2017]
**Paper Update**

We have updated the paper. For the updated paper, we re-ran the MNIST experiments with random hyperparameter search and now plot the log-loss of the test set in addition to the training set log-loss. We also added more detail about the synthetic experiment set-up. Last, we included a plot showing the Frobenius norm of the gradient as training progresses under different weight initializations.
Thank you all for your suggestions!

[Final Decision · Program Chairs · 06 Feb 2017]
**ICLR committee final decision**

This was a borderline paper. However, no reviewers were willing to champion the acceptance of the paper during the deliberation period. Furthermore, in practice, initialization itself is a hyperparameter that gets tuned automatically. To be a compelling empirical result, it would be useful for the paper to include a comparison between the proposed initialization and a tuned arbitrary initialization scale with various tuning budgets. Additionally, other issues with the empirical evaluation brought up by the reviewers were only partially resolved in the revisions. For these reasons, the paper has been recommended for rejection.